# Complementary, alternative, and integrative medicine researchers' practices and perceived barriers related to open science: An international, cross-sectional survey

Jeremy Y. Ng[1,2,3]*, Lucas J. Santoro[2,3], Kelly D. Cobey[4,5], Amie Steel[6], Holger Cramer[2,3], David Moher[1,5]

1 Ottawa Hospital Research Institute, Centre for Journalology, Ottawa, Canada, 2 Institute of General Practice and Interprofessional Care, University Hospital Tübingen, Tübingen, Germany, 3 Robert Bosch Center for Integrative Medicine and Health, Bosch Health Campus, Stuttgart, Germany, 4 Metaresearch and Open Science Program, University of Ottawa Heart Institute, Ottawa, Canada, 5 School of Epidemiology and Public Health, University of Ottawa, Ottawa, Canada, 6 Faculty of Health, Australian Research Consortium in Complementary and Integrative Medicine (ARCCIM), School of Public Health, University of Technology Sydney, Ultimo, Australia

* ngjy2@mcmaster.ca, jerng@ohri.ca

**Data Availability Statement:** All relevant data are included in this manuscript or posted on Open

## Abstract

### Introduction and objective

Open science (OS) aims to make the dissemination of knowledge and the research process transparent and accessible to everyone. With the increasing popularity of complementary, alternative, and integrative medicine (CAIM), our goal was to explore what are CAIM researchers' practices and perceived barriers related to OS.

### Methods

We conducted an anonymous online survey of researchers who published in journals listed in Scopus containing the words "complementary", "alternative", or "integrative" medicine in their names. We emailed 6040 researchers our purpose-built electronic survey after extracting their email address from one of their publications in our sample of journals. We questioned their familiarity with different OS concepts, along with their experiences and challenges engaging in these practices over the last 12 months.

### Results

The survey was completed by 392 researchers (6.5% response rate, 97.1% completion rate). Most respondents were CAIM researchers familiar with the overall concept of OS, indicated by those actively publishing open access (OA) (n = 244, 76.0%), registering a study protocol (n = 148, 48.0%), and using reporting guidelines (n = 181, 59.0%) in the past 12 months. Preprinting, sharing raw data, and sharing study materials were less popular. A lack of funding was reported as the greatest barrier to publishing OA by most respondents (n = 252, 79.0%), and that additional funding is the most significant incentive in applying

Science Framework: https://doi.org/10.17605/OSF.IO/AX6ZE.

**Funding:** The author(s) received no specific funding for this work.

**Competing interests:** The authors have declared that no competing interests exist.

**Abbreviations:** ASJC, All Science Journal Classification; CAIM, complementary, alternative, and integrative medicine; CHERRIES, Checklist for Reporting Results of Internet E-Surveys; FORRT, Framework for Open and Reproducible Research Training; OS, open science; OSF, Open Science Framework; PMID, PubMed identifier.

more OS practices to their research (n = 229,72.2%). With respect to preprinting barriers, 36.3% (n = 110) participants believed there are potential harms in sharing non-peer-reviewed work and 37.0% (n = 112) feared preprinting would reduce the likelihood of their manuscript being accepted by a journal. Respondents were also concerned about intellectual property control regarding sharing data (n = 94, 31.7%) and research study materials (n = 80, 28.7%).

## Conclusions

Although many participants were familiar with and practiced aspects of OS, many reported facing barriers relating to lack of funding to enable OS and perceived risks of revealing research ideas and data prior to publication. Future research should monitor the adoption and implementation of OS interventions in CAIM.

## Introduction

The goal of open science (OS) is to make the dissemination of knowledge and the research process faster, transparent, and open to everyone. Open access publishing, data and code sharing, and the distribution of open research resources are all examples of OS practices. Several jurisdictions in the research and publishing communities have adopted regulations and roadmaps to promote effective implementation and adoption of OS, while there is also growing momentum worldwide for OS practices to become more deeply ingrained in the research ecosystem [1–6]. Up to 85% of research may be wasted according to prior studies[6], with evidence indicating that the scientific system is exposed to problems such as publication bias, insufficient reporting standards, and lack of reproducibility [7–13]. By reducing avoidable study duplication, researchers and publishers can save both time and limited resources and funding. Moreover, making various stages of the research lifecycle accessible not only reduces bias, but also fosters creativity and ingenuity as anybody is free to use and build upon study data and resources. By limiting constraints on information access, the OS movement also supports equity. Ideally, this would lead to the removal of obstacles that the public and researchers encounter when it comes to accessing health information in the field of medicine, for example. However, no nation or research field has fully embraced OS despite the value of doing so on a global scale. There are a number of actual and perceived difficulties in deviating from the typical practice of 'closed research'. A number of problems have been mentioned as potential obstacles, such as how to successfully alter behavior to promote OS initiatives, how to educate academics on the formal procedures associated with OS, and how to balance openness with intellectual property protection [14–16].

Complementary, alternative, and integrative medicine (CAIM) has been a difficult field of medicine to define due to its dynamic nature. The US National Center for Complementary and Integrative Health (NCCIH) has differentiated between *complementary* and *alternative* medicine, with the former described as non-mainstream approaches used together with conventional medicine, and the latter described as non-mainstream approaches used instead of conventional medicine [17,18]. The NCCIH further describes integrative health as the convergence of conventional and complementary approaches in a coordinated way [17,18]. The most recent operational definition of CAIM was informed by a systematic search of quality-assessed information resources and includes 604 unique therapies [19]. Despite the increasing

popularity of CAIM therapies worldwide, many barriers remain with respect to the conduct and rigor of CAIM research [20]. To the best of our knowledge, no research has yet investigated what are CAIM researchers' practices and perceived barriers related to OS. To assess the state of OS in their communities, researchers in fields such as social science [21], economics [22], and psychology [23–25] have participated in extensive surveys. These surveys serve as the foundation for developing new interventions that more successfully implement open research, which can also be used to track the progression of OS using a longitudinal design. By gathering and understanding the practices and perceived barriers that CAIM researchers face concerning OS, we can then identify the best ways to increase OS adoption in the CAIM field.

We used a cross-sectional online survey, sent to authors who have been selected based on their status as corresponding authors on publications in journals indexed in Scopus that contain the words "complementary", "alternative" and/or "integrative" medicine in their journal titles. Our goal was to explore what are CAIM researchers' practices and perceived barriers related to OS. This is a descriptive study, with no formal hypothesis.

## Methods

### Ethics approval and consent to participate

We sought and were granted ethics approval by the Ottawa Health Science Network Research Ethics Board prior to beginning this project (OHSN REB Number: 20220642-01H). Implied consent was collected from each participant; upon clicking the survey invitation link, participants were presented with the consent form, and were informed that "By completing the survey your consent to participate is implied.".

### Transparency statement

Approval from the Ottawa Health Sciences Research Ethics Board was received before beginning this project (REB Number: 20220642-01H). Implied consent was collected from each participant; upon clicking the survey invitation link, participants were presented with the consent form, and were informed that "By completing the survey your consent to participate is implied.". Prior to participant recruitment, a study protocol was registered and made available on the Open Science Framework (OSF) [26]. The study materials, including survey, and de-identified raw data were shared using OSF at the time of the study's preprint being posted: https://doi.org/10.17605/OSF.IO/AX6ZE. A preprint of the study is also available on medRxiv at: https://doi.org/10.1101/2023.10.24.23297458.

### Study design

We conducted an anonymous, online, cross-sectional survey of a sample of authors who have published in CAIM journals indexed in Scopus from January 1, 2018 –December 31, 2022.

### Sampling framework

Journals containing the words "complementary", "alternative" or "integrative" in their names were chosen from Table 2 of Ng [27] which is based on the All Science Journal Classification (ASJC) pertaining to the Scopus category "complementary and alternative medicine" (code 2707) [28]. The full list of journals can be found in **Table 1**. All manuscripts published in these journals from January 1, 2018 –December 31, 2022, which had a corresponding author and associated PubMed identifier (PMID) were selected. The corresponding author's name and email address were then extracted from each PMID [29]. Authors who have published manuscripts of any type were included in this study [29]. Please refer to **Table 2** for a complete

**Table 1. List of journals from which author names and email addresses were extracted.**

| Source Title | Publisher | ISSN |
|---|---|---|
| Advances in Integrative Medicine | Elsevier | 2212–9588 |
| African Journal of Traditional, Complementary and Alternative Medicines | African Networks on Ethnomedicines | 0189–6016 |
| Alternative and Complementary Therapies | Mary Ann Liebert | 1076–2809 |
| Alternative Medicine | Future Medicine Ltd. | 1081–4000 |
| Alternative Medicine Alert | American Health Consultants, Inc. | 1096-942X |
| Alternative Medicine Review | Thorne Reasearch Inc. | 1089–5159 |
| Alternative Therapies in Health and Medicine | InnoVision Communications | 1078–6791 |
| Alternative Therapies in Women's Health | American Health Consultant | 1522–3396 |
| BMC Complementary and Alternative Medicine | Springer Nature | 1472–6882 |
| BMC Complementary Medicine and Therapies | Springer Nature | 2662–7671 |
| Chinese Journal of Integrative Medicine | Springer Nature | 1672–0415 |
| Complementary Health Practice Review | SAGE | 1533–2101 |
| Complementary Medical Research | Taylor & Francis | 0268–4055 |
| Complementary Medicine Research | Karger | 2504–2092 |
| Complementary Therapies in Clinical Practice | Elsevier | 1744–3881 |
| Complementary Therapies in Medicine | Elsevier | 0965–2299 |
| Complementary Therapies in Nursing and Midwifery | Elsevier | 1353–6117 |
| European Journal of Integrative Medicine | Elsevier | 1876–3820 |
| Evidence-based Complementary and Alternative Medicine | Hindawi | 1741-427X |
| Evidence-Based Integrative Medicine | Springer Nature | 1176–2330 |
| Focus on Alternative and Complementary Therapies | Wiley-Blackwell | 1465–3753 |
| Integrative Cancer Therapies | SAGE | 1534–7354 |
| Integrative Medicine | InnoVision Communications | 1546-993X |
| Integrative Medicine Alert | American Health Consultants, Inc. | 2325–2812 |
| Integrative Medicine Insights | Libertas Academica | 1177–3936 |
| Integrative Medicine Research | Elsevier | 2213–4220 |
| Journal of Alternative and Complementary Medicine | Mary Ann Liebert | 1075–5535 |

*(Continued)*

**Table 1.** (Continued)

| Source Title | Publisher | ISSN |
|---|---|---|
| Journal of Ayurveda and Integrative Medicine | Elsevier | 0975–9476 |
| Journal of Cancer Integrative Medicine | Prime National Publishing Corp. | 1544–6301 |
| Journal of Complementary and Integrative Medicine | Walter de Gruyter | 1553–3840 |
| Journal of Complementary Medicine | Australian Pharmaceutical Publishing Co., Ltd. | 1446–8263 |
| Journal of Evidence-Based Complementary and Alternative Medicine | SAGE | 2156–5872 |
| Journal of Evidence-Based Integrative Medicine | SAGE | 2515-690X |
| Journal of Experimental and Integrative Medicine | Gesdav | 1309–4572 |
| Journal of Integrative Medicine | Elsevier | 2095–4964 |
| Journal of the Society for Integrative Oncology | B.C. Decker Inc. | 1715-894X |
| Journal of Traditional and Complementary Medicine | Elsevier | 2225–4110 |
| Scientific Review of Alternative Medicine | Prometheus Books Inc. | 1095–0656 |
| Seminars in Preventive and Alternative Medicine | Elsevier | 1556–4061 |
| Traditional and Integrative Medicine | Tehran University of Medical Sciences | 2476–5104 |

explanation of our method for retrieving author emails adapted from Cobey et al [29]. A sample size calculation was not conducted since this is a convenience sample with descriptive work and the absence of any inferential testing.

**Table 2. Strategy for author name and email address retrieval.**

*Journal Search*
Journals containing the words "complementary", "alternative" or "integrative" in their names will be chosen from Table 2 of Ng[27]. This list contains journals belonging to the Scopus category "complementary and alternative medicine" (code 2707) which were identified based on the All Science Journal Classification.

*Scopus search strategy*
ISSN (22254110) OR ISSN (26627671) OR ISSN (20954964) OR ISSN (15347354) OR ISSN (09652299) OR ISSN (2515690X) OR ISSN (17443881) OR ISSN (16720415) OR ISSN (10755535) OR ISSN (22134220) OR ISSN (1741427X) OR ISSN (18763820) OR ISSN (09759476) OR ISSN (15533840) OR ISSN (25042092) OR ISSN (22129588) OR ISSN (10786791) OR ISSN (1546993X) OR ISSN (10762809) OR ISSN (24765104) OR ISSN (23252812) OR ISSN (01896016) OR ISSN (10814000) OR ISSN (1096942X) OR ISSN (10895159) OR ISSN (15223396) OR ISSN (14726882) OR ISSN (15332101) OR ISSN (02684055) OR ISSN (13536117) OR ISSN (11762330) OR ISSN (14653753) OR ISSN (11773936) OR ISSN (15446301) OR ISSN (14468263) OR ISSN (21565872) OR ISSN (13094572) OR ISSN (1715894X) OR ISSN (10950656) OR ISSN (15564061)

*Article Retrieval*
We will search for all articles published in each journal using the ISSN number of included journals.
We will run search for each journal separately. After each search, we will sort the results by Entry date (descending) and export all publications from January 1, 2018 to December 31, 2022.

*Email Retrieval*
The list of PMID numbers will be exported as an.csv file and input into an R script (built based on the easyPubMed package) to retrieve the authors' name, affiliation institutions and email addresses.

### Participant recruitment

Only researchers who were identified using our sampling framework (**Table 2**) were contacted to take part in our study and complete the closed survey. SurveyMonkey was used to send emails to the authors captured in our sample. The prospective participants received an email containing an explanation of the study and its goals on February 12, 2023. This email also contained a link to an informed consent form which participants had to agree to before they could access the online survey. Reminder emails were sent to participants after the first, second, and third weeks after the original invitation email. The survey closed two weeks after the final reminder email on March 19, 2023. There was no financial compensation and no requirement to participate in this study. Any participant who did not wish to respond to a question could skip it.

### Survey design

The complete survey adapted from Cobey et al [29] can be found on OSF. It contained 34 items in total, and was displayed across 12 pages (screens). The survey began by asking participants a screening question followed by six general demographic questions (e.g., location, age). They were then asked five questions regarding their role in research and their expertise. Using the Framework for Open and Reproducible Research Training (FORRT) Glossary [30], several definitions of OS and OS practices were presented to participants, of which they were asked to indicate how familiar they were with each concept. The remaining groups of questions asked participants about their experiences engaging in OS practices in the past 12 months and the barriers they encountered. There were 32 multiple-choice questions and 2 open-ended questions. All questions were optional and could be skipped to proceed through the survey. The survey was pilot tested by two independent CAIM researchers to integrate their feedback into the survey prior to distribution.

### Data management, analysis, and reporting

The survey data that was gathered was exported to Microsoft Excel. Basic descriptive statistics including counts and percentages were generated based on the analysis of the quantitative data. With respect to qualitative data, a thematic content analysis was conducted by two researchers who analyzed repeated ideas from the open-ended text-based responses. The researchers first collected, categorized, and grouped the responses into themes separately, followed by a discussion to obtain a final consensus of the themes and codes for reporting. The Checklist for Reporting Results of Internet E-Surveys (CHERRIES) was used to report results [31].

## Results

### Search strategy

In total, 40 journals were eligible according to our search criteria. After searching these journals for publications between January 1, 2018 and December 31, 2022, 16 175 articles with an accompanying PMID were extracted. From those articles, 6040 unique email addresses were obtained as not all articles contained an extractable corresponding author email address. Our survey's raw, deidentified data is available here: https://osf.io/ytxqd.

### Demographics

Overall, 392 researchers participated in our survey out of the 6040 authors emailed (6.5% response rate, 97.1% completion rate); it should be noted that not all participants responded to

all questions, hence we provide the total number of responses for each of the questions in parentheses. Responses were deemed incomplete when no questions were answered after the initial screening question. Moreover, we have reported our raw response rate, which is under-estimated as we cannot determine how many of the 6040 authors who were emailed currently identify as a CAIM researcher. It should be noted that we also did not calculate email bounce back rate, although we assume this to be around 10%, hence the response rate of those who were invited is in fact slightly higher. The survey took approximately 24 minutes to complete. The majority of respondents (n = 375, 95.7%) answered "yes" when asked if they identify as a researcher of CAIM therapies. Respondents were primarily located in South-East Asia (n = 99, 27.2%), Americas (n = 73, 20.2%), or Europe (n = 72, 19.9%), and identified as a senior career researcher (n = 186, 53.1%) holding positions of faculty member/principal investigator (n = 235, 67.1%). Most respondents also indicated that CAIM was their primary area of research (n = 218, 62.6%), with 47.1% (n = 165) of participants focusing on clinical research. Complete participant demographics are available in **Table 3**. Furthermore, crosstabs by age, career stage, whether they are a caregiver, gender, whether they belong to a minority group, and WHO region are provided on OSF: https://osf.io/ax6ze/.

## Open science practices and experiences

The respondents were "somewhat familiar" (n = 69, 21.7%) with the overall concept of OS, with more than half of respondents cumulatively being "very familiar" (n = 82, 25.8%) or "moderately familiar" (n = 132, 41.5%). Only 2.83% of respondents (n = 9) reported being "not at all familiar" with the concept of OS (**Fig 1**). After asking participants their familiarity with various OS practices, respondents were "very familiar" with open access publishing (n = 210, 64.2%), preprinting (n = 134, 41.7%), reporting guidelines (n = 141, 44.3%), and protocol registration (n = 122, 38.2%). Respondents also reported being "moderately familiar" with the practices of open data (n = 118, 37.0%) and open materials (n = 98, 31.0%). While almost 80% of respondents were at least "slightly familiar" with the practice of patient and public involvement in research, 10.6% (n = 34) reported being "not at all familiar", the highest among the seven open science practices investigated (**Fig 2**).

## Open science training, promotion, and implementation

When asked the source of their training or knowledge about OS, of the 317 respondents who answered, more than half reported learning or training on their own accord while conducting research (n = 188, 59.5%). Approximately one in three participants received no formal training with respect to OS (n = 104, 32.9%), followed by 21.8% (n = 69) receiving mentorship from their supervisor/peers, with 18.0% (n = 57) participating in formal coursework or workshops about OS (**Fig 3**). When probed about their preferred format of OS training, the majority of respondents ranked a website of resources (not including webinars) as their first choice (n = 128, 43.1%). In-person lectures (n = 136, 46.0%) and in-person workshops (n = 131, 44.1%) were the least preferred formats, being ranked by most respondents 6th and 7th respectively (**Fig 4**). Almost three quarters of respondents indicated that additional funding would be the greatest incentive to allow respondents to apply more OS practices to their research (n = 229, 72.2%). About half of respondents reported that practical support from their institution (e.g., a person to ask questions about the practicalities of performing OS) would also be helpful (n = 161, 50.8%), followed by clearer communication about the benefits and value that OS has to research (n = 145, 45.7%). Approximately one third of participants noted that additional training on applying OS practices would also incentivize their application (n = 105, 33.1%). Roughly a quarter of respondents (n = 83, 26.2%) expressed that being recognized for

**Table 3. Characteristics of survey participants.**

| **Sex (n = 363)** | |
|---|---|
| Male | 166 (45.7%) |
| Female | 189 (52%) |
| Other | 8 (2.2%) |
| **Age (n = 363)** | |
| 25–34 | 52 (14.3%) |
| 35–44 | 124 (34.2%) |
| 45–54 | 102 (28.1%) |
| 55–64 | 68 (18.7%) |
| >65 | 16 (4.4%) |
| Prefer not to say | 1 (0.3%) |
| **Visible Minority (n = 363)** | |
| Yes | 66 (18.2%) |
| No | 276 (76%) |
| Prefer not to say | 21 (5.8%) |
| **Disability (n = 364)** | |
| Yes | 15 (4.1%) |
| No | 344 (94.6%) |
| Prefer not to say | 5 (1.4%) |
| **Caregiver (n = 362)** | |
| Yes | 158 (43.7%) |
| No | 196 (54.1%) |
| Prefer not to say | 8 (2.2%) |
| **Location (n = 361)** | |
| Africa | 28 (7.8%) |
| Americas | 73 (20.2%) |
| Eastern Mediterranean | 33 (9.1%) |
| Europe | 72 (19.9%) |
| South-East Asia | 99 (27.4%) |
| Western Pacific | 56 (15.5%) |
| **Current Position (n = 350)** | |
| Graduate Student | 14 (4%) |
| Postdoctoral fellow | 27 (7.7%) |
| Faculty member/principal investigator | 235 (67.1%) |
| Research support staff | 18 (5.1%) |
| Scientist in industry | 9 (2.6%) |
| Scientist in third sector | 6 (1.7%) |
| Government scientist | 16 (4.6%) |
| Other | 25 (7.1%) |
| **Career Stage (n = 350)** | |
| Graduate student | 7 (2%) |
| Early career researcher (<5 years post education) | 52 (14.9%) |
| Mid-career research (5–10 years post education) | 94 (26.9%) |
| Senior researcher (>10 years post education) | 186 (53.1%) |
| Other | 11 (3.1%) |
| **Primary Research Area (n = 350)** | |
| Clinical research | 165 (47.1%) |
| Preclinical research (in vivo) | 38 (10.9%) |

(*Continued*)

**Table 3.** (Continued)

| | |
|---|---|
| Preclinical research (in vitro) | 28 (8%) |
| Health systems research | 18 (5.1%) |
| Health services research | 46 (13.1%) |
| Methods research | 22 (6.3%) |
| Epidemiological research | 15 (4.3%) |
| Other | 18 (5.1%) |
| **CAIM Research Priority (n = 348)** | |
| Primary (most research about CAIM) | 218 (62.7%) |
| Secondary (most research not about CAIM) | 130 (37.4%) |
| **Area of CAIM Research (n = 350)** | |
| Mind-body therapies | 91 (26%) |
| Biologically based practices | 116 (33.1%) |
| Manipulative and body-based practices | 59 (16.9%) |
| Biofield therapy | 15 (4.3%) |
| Whole medical systems | 130 (37.1%) |
| Other | 58 (16.6%) |

applying OS practices with respect to being hired, promoted and tenured would be valuable, with another quarter (n = 80, 25.2%) indicating that having additional staff trained on OS practices would allow them to apply more OS practices to their research (**Fig 5**). Respondents closely ranked funders (n = 100, 37.0%) and research institutions (n = 97, 35.9%) as the most significant stakeholders to create policies resulting in the successful uptake of OS. Scholarly journals and scholarly societies were ranked lower, with 60.9% of respondents (n = 165) ranking the latter as the least significant stakeholder (**Fig 6**). When 163 participants responded to the open-ended question asking about the best ways to promote open science, 205 codes were generated. The 205 codes were grouped into 27 subthemes, and then into 6 overarching

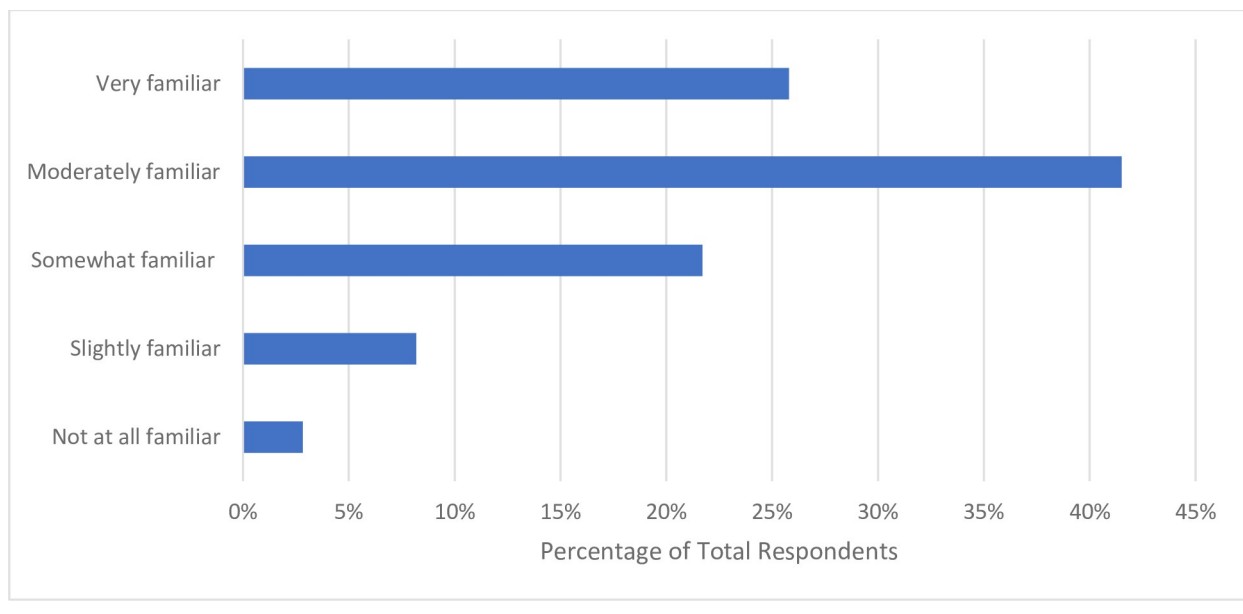

**Fig 1. Participant familiarity with open science in general.**

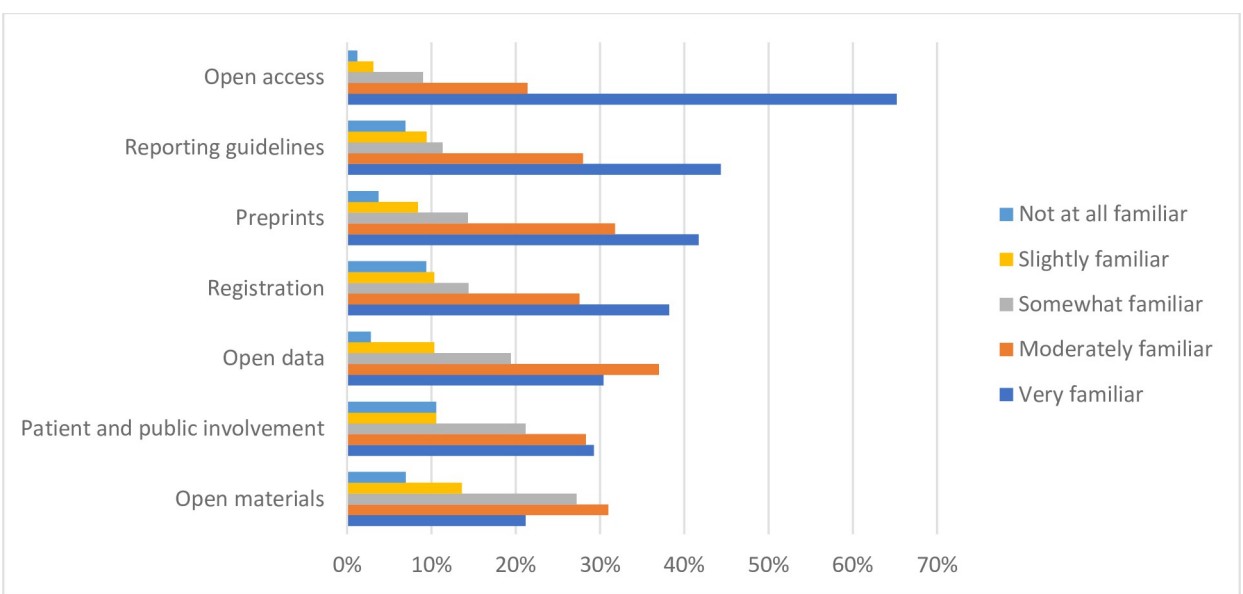

**Fig 2. Participant familiarity with various open science practices.**

themes. Some prevalent subthemes that emerged were, reducing or eliminating expensive article processing charges (APCs) when submitting manuscripts to open access journals, increasing financial support from funders and institutions and promoting OS: https://osf.io/ex7ky (**Table 4**). When 82 respondents shared any thoughts or opinions at the end of the survey, 26 codes were generated which were grouped into 16 subthemes, and then into 7 overarching themes. Prevalent subthemes that emerged were that the costs of OS are too high, and that funding is mandatory for OS: https://osf.io/xpgft (**Table 5**).

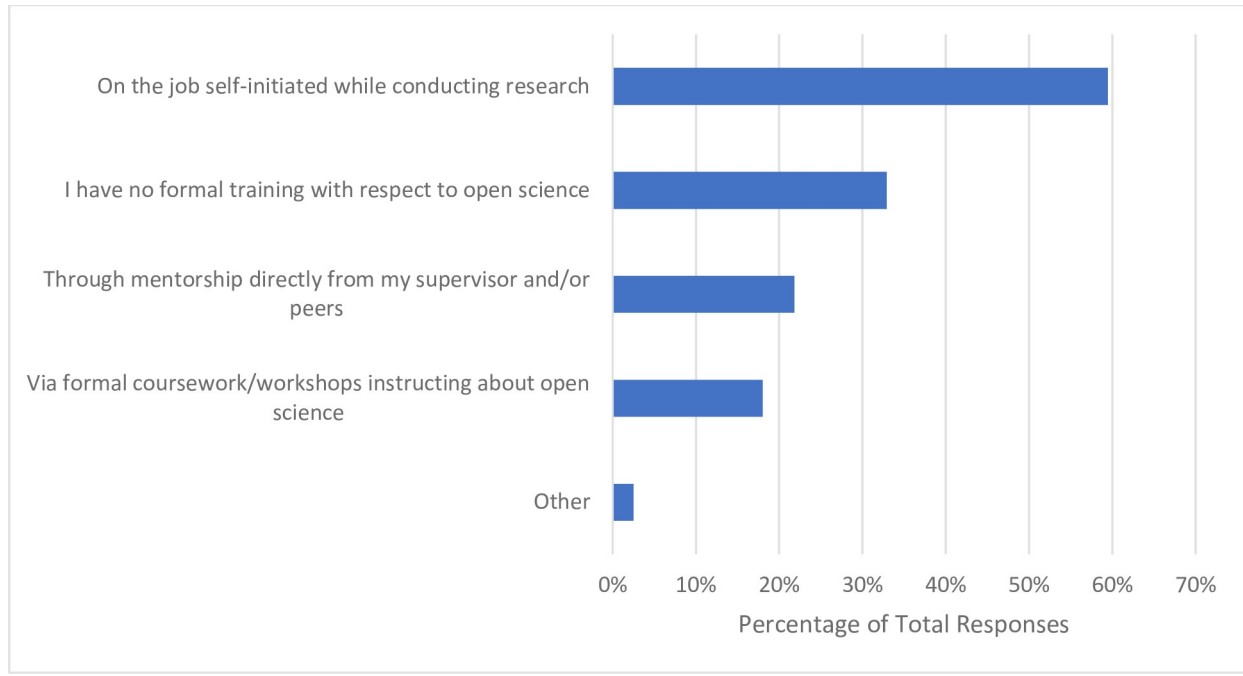

**Fig 3. Where participants have received training or learned about open science from.**

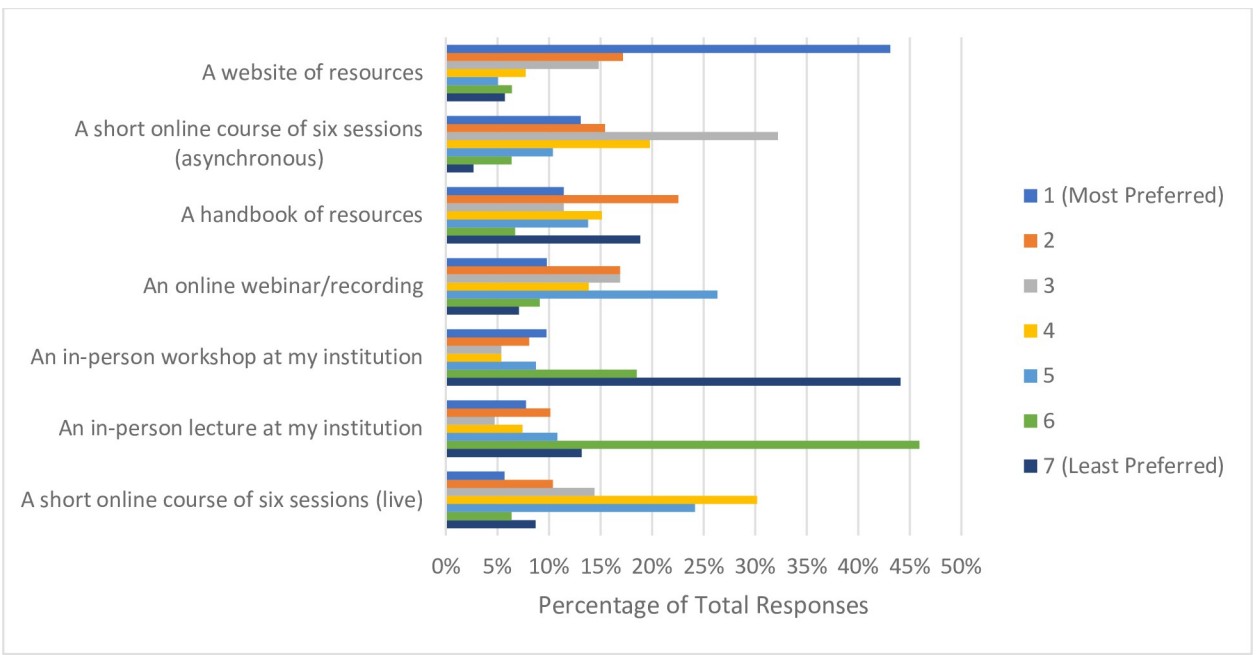

**Fig 4. Ranked formats of open science training participants would prefer.**

## Open access

In the past 12 months, 76.0% of respondents (n = 244) indicated they have published an article 'open access' (**Table 6**). When asking respondents to identify the most prominent barriers preventing them from publishing open access are, roughly four out of five respondents (n = 252, 79.0%) reported insufficient financial capital to cover high APCs that are common with open

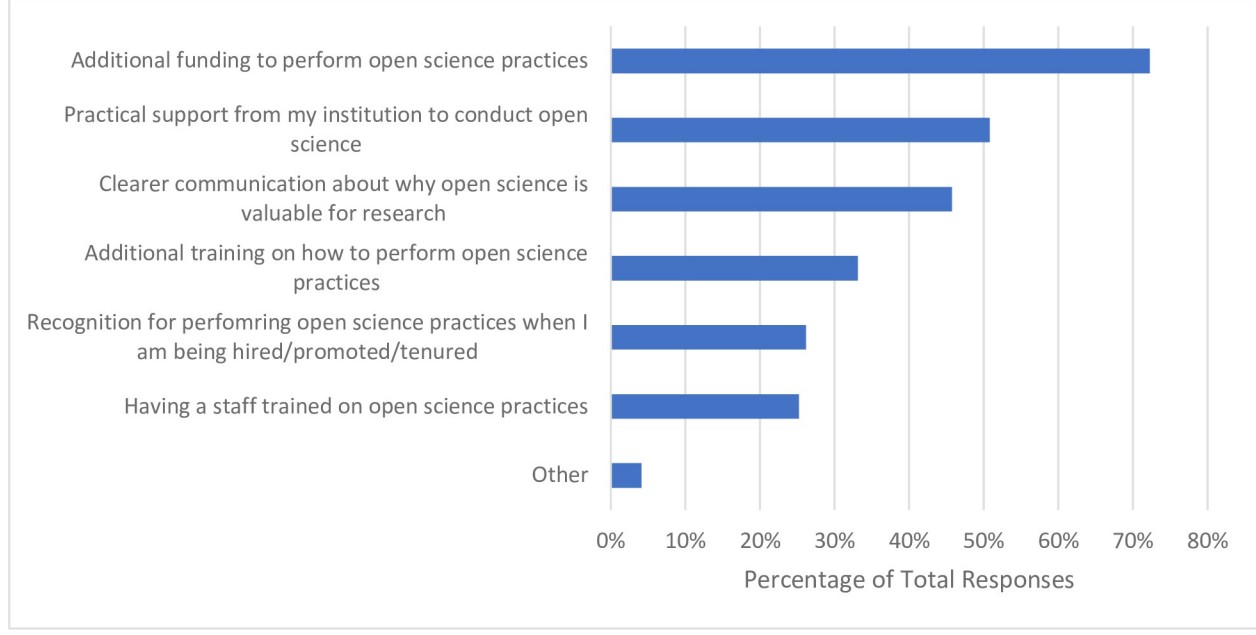

**Fig 5. Incentives participants feel would allow for them to apply more open science practices to their research.**

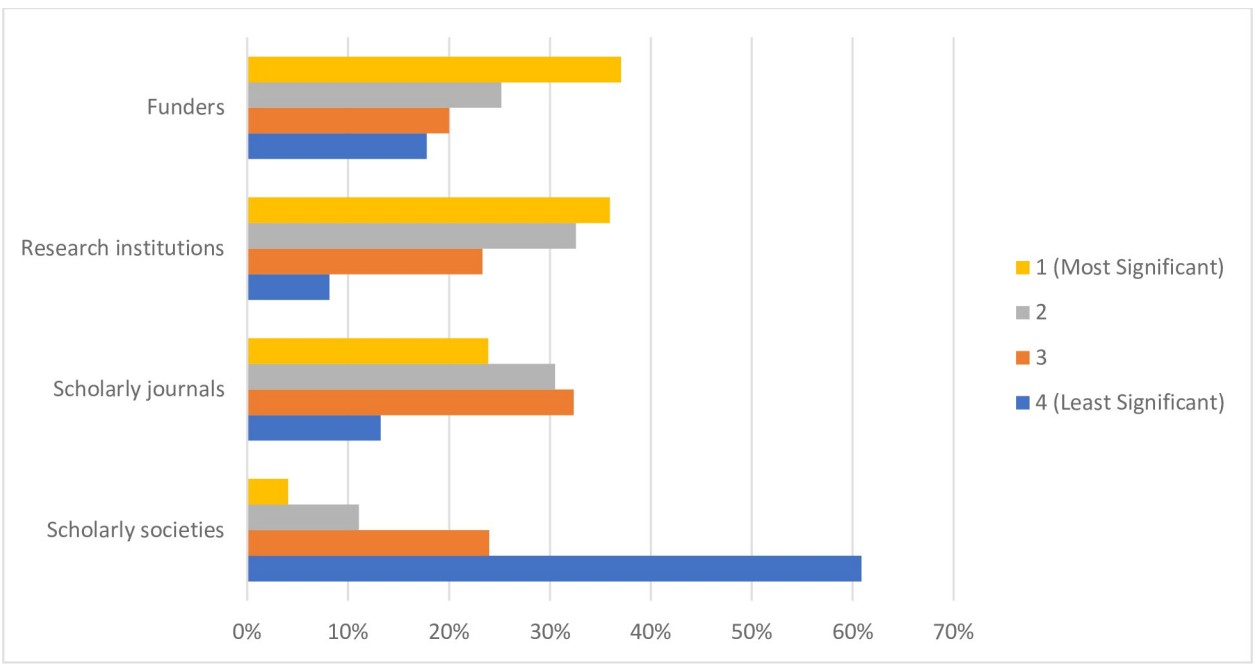

**Fig 6. Stakeholders ranked based on their impact to create policies resulting in the successful uptake of open science.**

access journals. A further 16.6% of respondents (n = 53) stated they believe their institution does not value the practice of publishing open access. Additionally, 10.3% of respondents (n = 33) reported not facing any of the perceived barriers mentioned in the survey (**Fig 7**).

## Preprinting

With respect to preprinting in the past 12 months, more than half of respondents (n = 194, 61.0%) indicated they have not produced a preprint prior to publishing an article (**Table 6**). Respondents were asked about the barriers they have faced with respect to creating preprints. Approximately one-third of respondents (n = 112, 37.0%) stated that they worry preprinting their work will reduce their manuscript's chance of being accepted by a peer reviewed journal, with another third of respondents indicating that they feel there are potential harms in sharing work that has not been peer reviewed (n = 110, 36.0%). Additionally, many respondents stated they do not see the benefit in making a preprint (n = 96, 31.7%), feel that their institution does not value creating preprints (n = 78, 25.7%), or that they do not know how to make a preprint (n = 71, 23.4%) (**Fig 8**).

## Sharing data

When participants were asked if they have shared the raw data for a research study at the time of publication, only 22.9% (n = 72) reported that they did in the past 12 months (**Table 6**). Respondents were also asked to identify barriers they have encountered with respect to sharing the raw data from their research when publishing. Participants principally selected that they have concerns regarding the unintended use of secondary data (n = 110, 37.0%), concerns about the misinterpretation of the data (n = 95, 32.0%), concerns of intellectual property control (n = 94, 31.7%), and concerns about patient privacy when data is shared (n = 86, 29.0%). Roughly one-quarter of respondents (n = 77, 25.9%) indicated that they do not know how to prepare their data appropriately for sharing (**Fig 9**).

**Table 4. Thematic content analysis of best way to promote open science.**

| Themes | Subthemes | Example | Number of Responses |
|---|---|---|---|
| **Advertising** | Promote OS | "Awareness campaigns." | 23 |
| | Open access promotion when submitting manuscript | "Current practice promoting open access selection upon submission/ registration of the manuscripts is good already." | 3 |
| | Promote reputable OS journals | "The journal society should change from a closed approach to an open-access." | 3 |
| | No additional promotion needed | "Current practice promoting open access selection upon submission/ registration of the manuscripts is good already." | 1 |
| **Outreach and Training** | Online webinars | "Online Webinars" | 1 |
| | OS training | ". . .educate new scientists in the area." | 20 |
| | Improve OS knowledge | "Improving knowledge in this field." | 12 |
| | Social media | "Social media." | 3 |
| **Finances** | Reduce APCs | "Make open science truly open by getting rid of fees." | 33 |
| | Additional funding | "Funding to publish open access." | 44 |
| | Pay peer reviewers | ". . .Pay peer-reviewers." | 2 |
| | Nonprofit publishing | "Change the for-profit model of publishing towards a not-for-profit model. . ." | 1 |
| **Policy** | Implement OS policy | "Issuing some mandatory rules from the government level and journals level." | 11 |
| | Increase OS incentives | "A financial model that rewards activities associated with open science." | 11 |
| | Include OS in HPT | ". . .Make it part of tenure tracks." | 2 |
| | Make OS mandatory | "Funders making it compulsory." | 15 |
| **OS Practices and Methodologies** | More rigorous OS peer review | "Rigor in reviewing process and trustworthiness in published papers." | 4 |
| | Increase open access articles | "Make availability of journal articles free, irrespective of funding." | 7 |
| | Include more OS practices in research | "Simply by doing it and being back up financially." | 6 |
| | Encourage open peer review | "Encouraging open peer review. . ." | 1 |
| | Develop open source software | "Developing open-source software. . ." | 1 |
| | Create reproducible code | ". . .creating reproducible code from the undergraduate level." | 1 |

*(Continued)*

**Table 4.** (Continued)

| Themes | Subthemes | Example | Number of Responses |
|---|---|---|---|
| **Other** | Be accepting to new ideas | "Accepting new ideas. . ." | 1 |
| | Collaborative research | "Collaborative research." | 7 |
| | Recognition by others | "Get recognized by own institutions and research communities. . ." | 7 |
| | Reduce research competition | "Reduce the need for competition in research. . ." | 1 |
| | Increase peer review speed | ". . .with reasonable timelines, to achieve open access." | 1 |

## Sharing study materials

Approximately two-thirds of respondents (n = 209, 67.6%) stated that they have not shared the study materials underpinning a study at the time of publication ([Table 6]). When asked which barriers they have encountered with regards to sharing study materials when publishing, greater than a quarter of respondents (n = 80, 28.7%) indicated having concerns about intellectual property control. Respondents also indicated that they have concerns about the

**Table 5. Thematic content analysis of participants' end of survey thoughts and comments.**

| Themes | Subthemes | Example | Number of Responses |
|---|---|---|---|
| **Finances** | Costs of open science are too high | "Open science carries a relatively high cost. . ." | 6 |
| | Costs of open science can be used elsewhere | ". . .these costs would rather go to purchase other requirements in the lab and the work gets published in the normal way." | 1 |
| | Funding is mandatory for open science | "Funding is mandatory for open science." | 3 |
| **CAIM** | Difficult to publish in the CAM field in general | ". . .It continues to be extremely difficult and political to publish in this area." | 1 |
| | Need to understand how CAM-related issues hinders open access | "Understanding the specific issues about CAM that may hinder open access is important." | 1 |
| **Reporting guidelines** | Reporting guidelines improve research transparency clarity and reproducibility | "I believe that the use of guidelines is useful to improve the transparency, clarity and reproducibility of the data of how the research was conducted." | 1 |
| | Stringent reporting guidelines | "Guidelines for Ayurvedic medicine are very stringent. . ." | 1 |
| **Open science implementation** | Need more people to agree to use open science practices | "Engagements are necessary." | 2 |
| | Need an open access implementation timeline | "An appropriate timeline for effective implementation of OA has not been established. . ." | 1 |
| **Research methodology** | Patient and public involvement in important | "I value patient and public involvement." | 2 |
| | Recruiting participants is difficult | "Recruiting participants remains problematic. . ." | 1 |
| | Use whole system research in health care | "It is important to my research and with research colleges to use whole system research in order to develop new and innovative health care services." | 1 |
| **Policy** | Peer review should be transparent | "Please make peer review process also transparent. . ." | 1 |
| | Implementing policies may do more harm than good | "I fear the outcome will be another policy that does more to prevent what you want. . ." | 1 |
| **Other** | Few incentives for new knowledge | "In my region there are few incentives for the generation of new knowledge." | 1 |
| | Inequality in science | ". . .however, it's a big multimillion business that increases the gap between the highly funded research and those who do not receive consistent or sufficient funds. . ." | 2 |

**Table 6. Participants' engagement in various open science practices in the past 12 months.**

| | Engaged in Open Science Practice | | | |
|---|---|---|---|---|
| | **Yes** | **No** | **Have not Published a Paper or Conducted a Study in the Past 12 Months** | **Do not know** |
| **Publish Open access (n = 321)** | 244 (76%) | 54 (16.9%) | 21 (6.6%) | 2 (0.6%) |
| **Preprint (n = 318)** | 100 (31.5%) | 194 (61%) | 19 (6%) | 5 (1.6%) |
| **Share Raw Data (n = 314)** | 72 (22.9%) | 224 (71.3%) | 15 (4.8%) | 3 (1%) |
| **Share Study Materials (n = 309)** | 74 (24%) | 209 (67.6%) | 16 (5.2%) | 10 (3.2%) |
| **Register a Study Protocol (n = 306)** | 148 (48.4%) | 133 (43.5%) | 22 (7.2%) | 3 (1%) |
| **Reference a Reporting Guideline Checklist (n = 306)** | 181 (59.2%) | 100 (32.7%) | 17 (5.6%) | 8 (2.6%) |

unintended use of their study materials (n = 73, 26.0%), they do not know how to prepare their study materials for sharing with the research community (n = 71, 25.5%), and they have concerns about the misinterpretation of their study materials (n = 64, 22.9%) (**Fig 10**).

## Registering a study protocol

Respondents were relatively split when asked if they have registered a study protocol in the past 12 months. Roughly half of respondents (n = 148, 48.4%) indicated they have registered a protocol in the past 12 months (**Table 6**). When asked what barriers they have faced with respect to registering their study protocol prior to starting a research project, roughly one fifth of respondents (n = 47, 20.4%) stated they do not know how to create a study registration. Respondents also indicated that they feared they will be scooped if they share their study plan

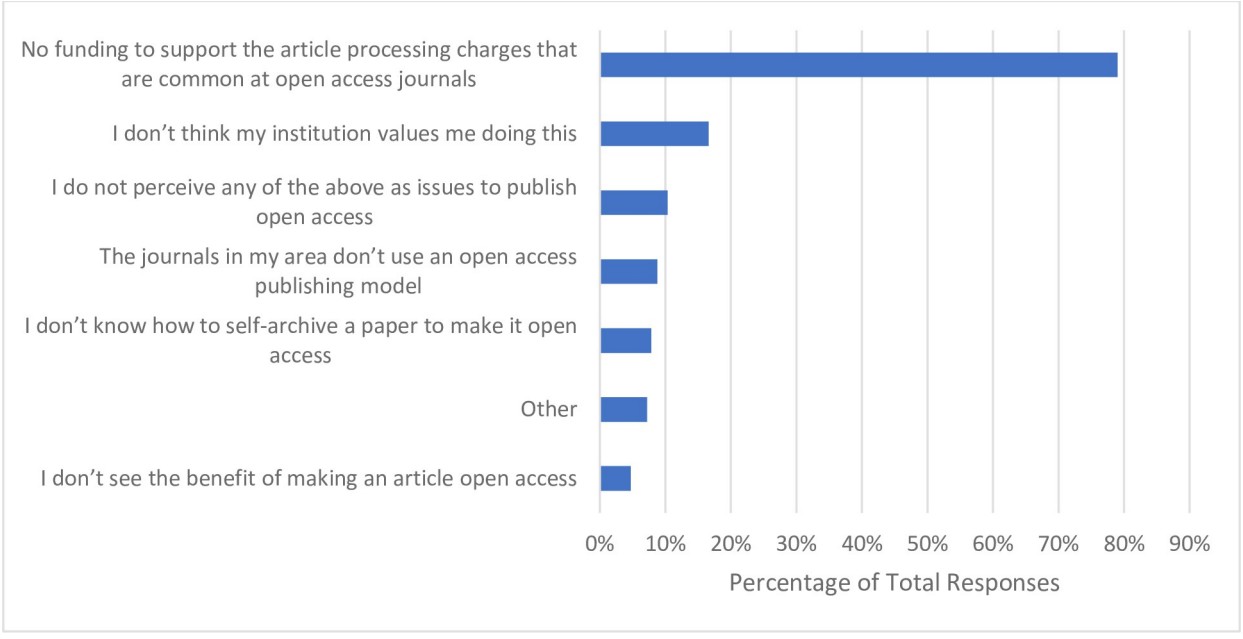

**Fig 7. Barriers participants have faced with respect to publishing open access.**

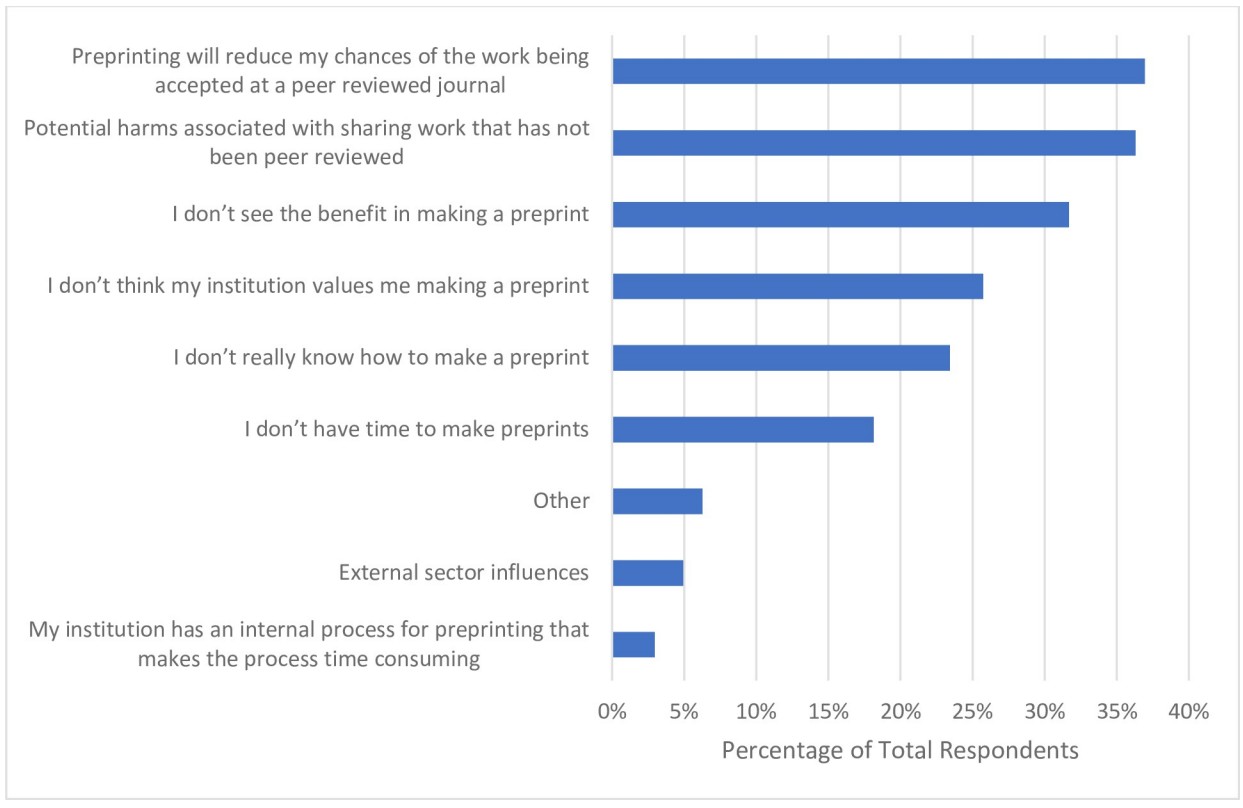

**Fig 8. Barriers participants have faced with respect to creating preprints.**

before publication (n = 45, 19.5%), while roughly another fifth (n = 44, 19.1%) stated that they do not have the time to register their studies (**Fig 11**).

## Using reporting guidelines

When participants were asked if they have used and referenced any reporting guideline checklists in the past 12 months, more than half of respondents (n = 181, 59.2%) said that they have (**Table 6**). Additionally, participants were asked to identify the barriers they encountered with respect to using reporting guidelines when reporting their research. Of the 212 respondents, 58 (27.4%) stated they do not know where to find relevant reporting guidelines. Respondents also indicated that they do not have the time to use reporting guidelines (n = 37, 17.5%), and that they do not know how to use reporting guidelines (n = 35, 16.5%) (**Fig 12**).

## Discussion

The goal of this study was to explore what are CAIM researchers' practices and perceived barriers related to OS. Our findings demonstrate that CAIM researchers are familiar with and actively publish open access, register study protocols and use reporting guidelines. However, CAIM researchers identify a lack of funding as a prominent barrier preventing further implementation of OS practices in their work, particularly concerning APCs when publishing open access. Additionally, CAIM researchers have concerns regarding intellectual property control and how their research is used/interpreted when sharing data, study materials, and non-peer reviewed work prior to publication. Overall, respondents from our study as well tend to be most familiar with the practice of open access publishing (n = 210, 64.2%) as this may be the

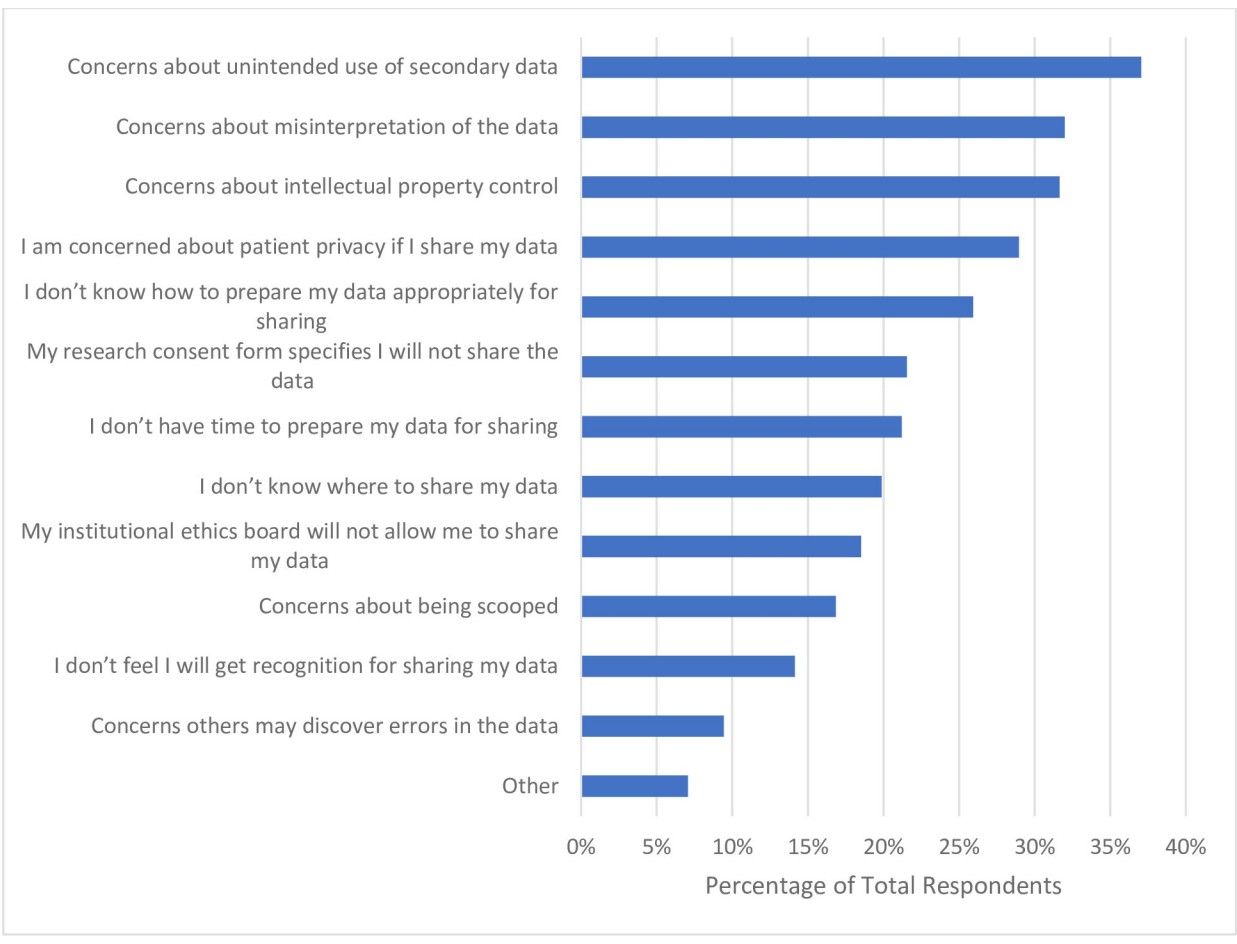

**Fig 9. Barriers participants have faced with respect to sharing raw data from research when publishing.**

most popular OS practice. It is estimated that roughly a third to almost 50% of published scientific research is available as open access [32]. These results are consistent with previous studies investigating researchers in various disciplines such as social sciences and humanities, engineering, and natural sciences which found that participants were most knowledgeable about open access publishing [33–36].

In addition, the present study found that many researchers were self-taught regarding OS practices while conducting research of their own (n = 188, 59.5%). This coincides with the fact that many participants preferred to learn about OS using a website of resources (n = 128, 43.1%). As this intervention is primarily self-initiated, completed alone, and for low cost, this may explain why our sample of researchers preferred it over in-person lectures or workshops, which they ranked as their least preferred formats. Altogether, our sample of CAIM researchers appear to lack formal training of OS practices. This is troubling as roughly half of our respondents (n = 161, 50.8%) indicated that practical support from their institution, along with clearer communication regarding the value of OS (n = 145, 45.7%) would incentivize them to apply more OS practices to their research. While greater institutional support would likely benefit CAIM researchers, much of CAIM training has been reported to take place in private colleges rather than universities, leading to limited placements and opportunities for further post-graduate education [20]. Therefore, CAIM researchers may not obtain the same

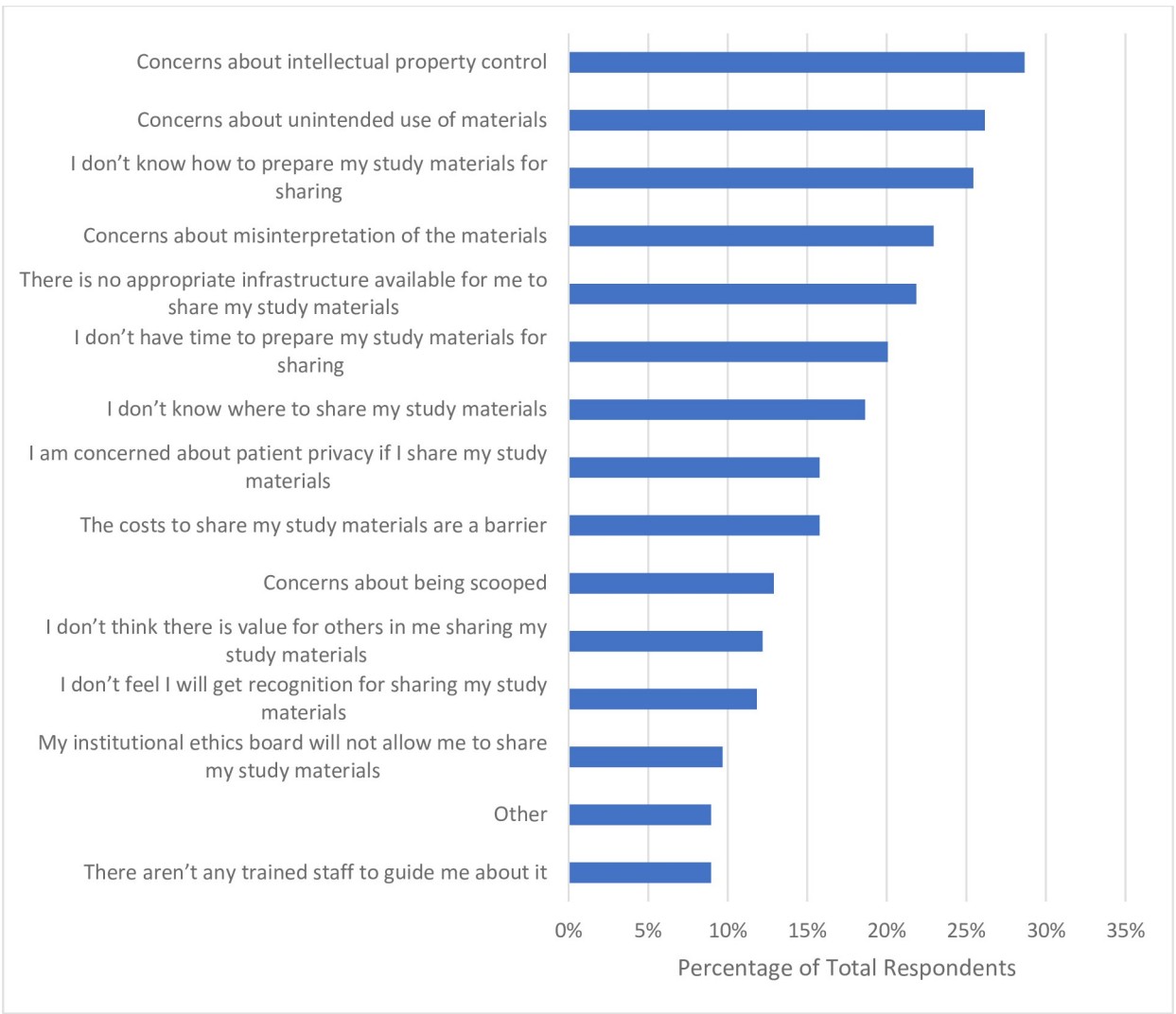

**Fig 10. Barriers participants have faced with respect to sharing study materials from research when publishing.**

research experiences as those in the mainstream biomedical field, leading to poorer research practices such as not incorporating international research reporting standards in their work [20]. Strategies such as educational training programs, increased CAIM-related masters and doctoral programs, and OS training would likely improve research literacy and evidence-based CAIM [37,38].

Approximately three quarters of respondents (n = 229, 72.2%) indicated that additional funding would be the greatest incentive to incorporate more OS practices into their research. This point was reiterated in the analysis of open text responses which identified participant preferences for financially motivated incentives such as additional funding or financial support, and waivers or discounts to APCs. One of the largest financial burdens faced by researchers include APCs and open access charges that are often very high and unaffordable, especially for researchers from developing countries [39]. For example, *BMC Complementary Medicine and Therapies* charges £1990.00/$2690.00/€2290.00 for each article accepted for publication [40]. This may explain why when asked to identify barriers to publish open access, three

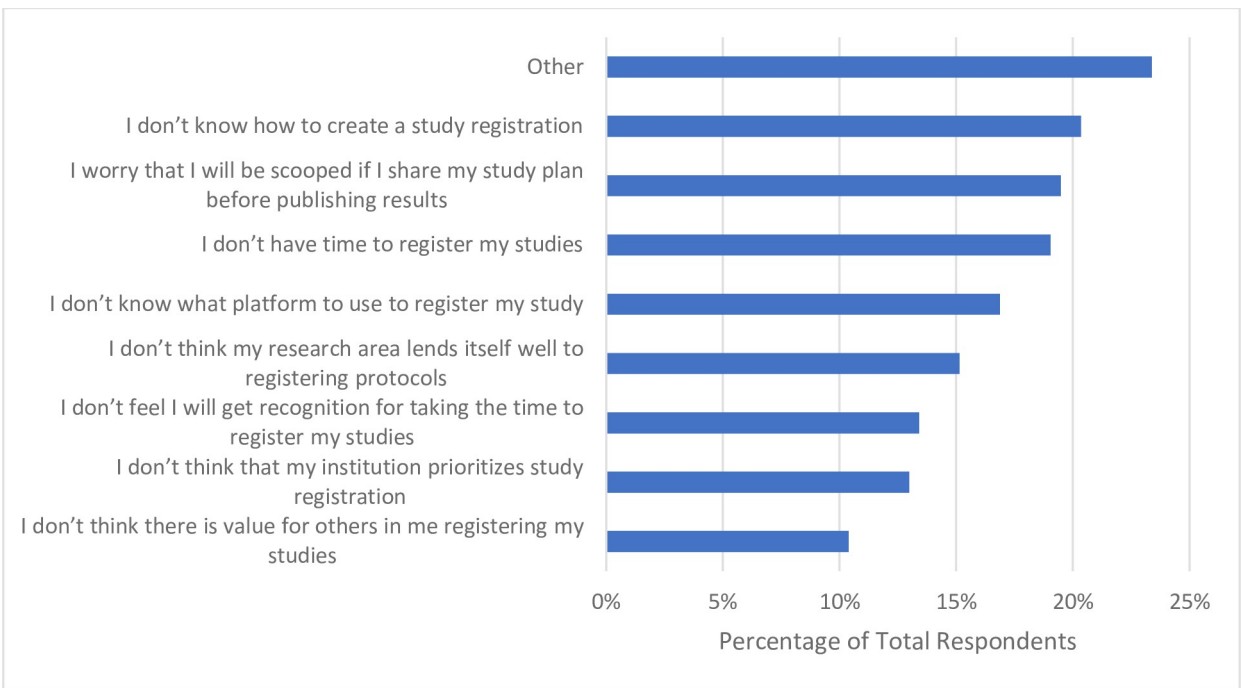

**Fig 11. Barriers participants have faced with respect to registering their study protocol prior to starting a research project.**

quarters of respondents (n = 252, 79.0%) indicated they do not have the funding to afford the APCs that are common at open access journals, despite a similar proportion of respondents (n = 244, 76.0%) having published open access in the past 12 months. Comparative studies such as the survey of Molecular Biology Society of Japan members found that 76.6% (n = 478)

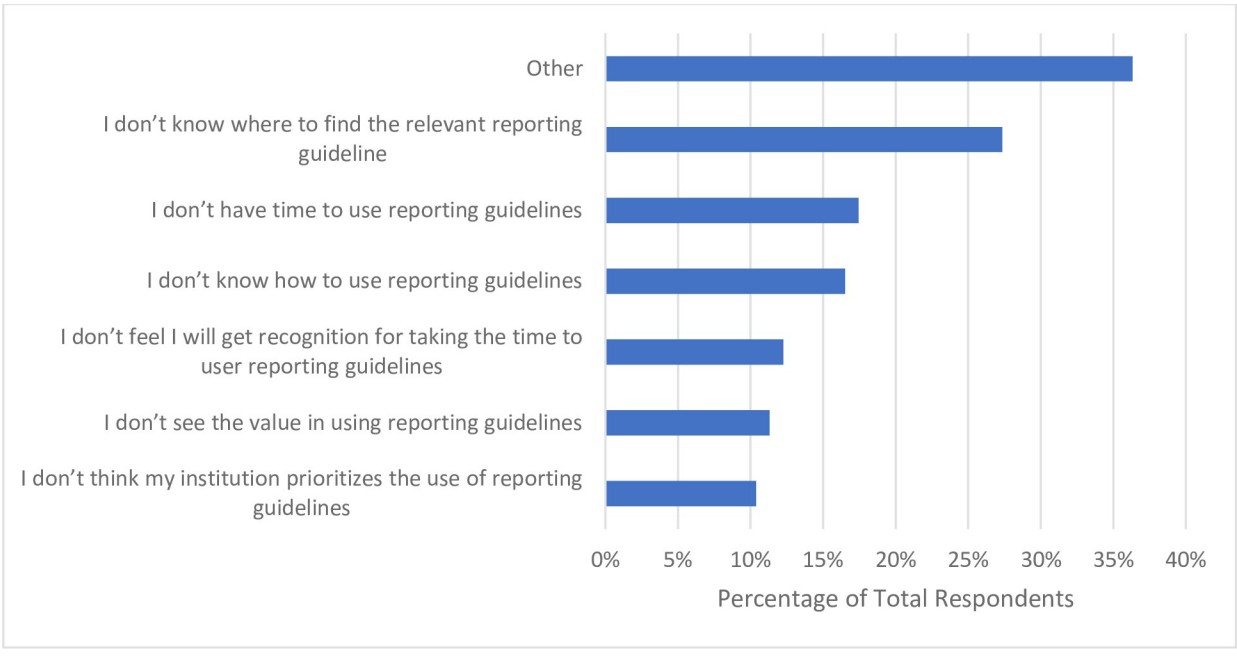

**Fig 12. Barriers participants have faced with respect to using reporting guidelines when reporting their research.**

respondents have published open access [35,41,42]. Additionally, 91.5% (n = 571) of respondents in the same survey reported wanting to publish open access [35]. Costs, however, remain a barrier to those wishing to publish open access, with some researchers having to use their personal funds towards APCs [35]. Ideally, OS practices such as open access publishing promote equity and reduce barriers that end-users and authors face. Tools such as Sherpa Romeo exist to help users understand individual journal policies regarding preprinting, post-printing, and open access publishing. Finding journals that lack APCs or imply charges should therefore be easier for readers and authors, which is especially important for those in lower-income countries.

A lack of understanding or knowledge regarding the OS practices of preprinting and data and materials sharing appears to be a theme among our sample of CAIM researchers. These were the least popular OS practices with similar barriers being reported between them. For example, 37.0% of participants (n = 112) report they worry preprinting will reduce the chance of their manuscript being accepted at a peer-reviewed journal despite greater than 60% of preprints posted before 2017 eventually being published [43]. With respect to sharing data and study materials, participants were concerned about intellectual property control, unintended use of their data, and data misinterpretation. However, researchers can quell these fears by actively publishing detailed metadata or descriptions of the data collection process/analysis, set permissions for data access and reuse, understandings the legal terms and conditions that exist to protect researcher's rights, and preprinting to make a first claim [44]. Overall, CAIM researchers would likely benefit from further training and experience regarding the incentives and benefits of various OS practices.

Registering a study protocol and referencing a reporting guideline checklist were more popular OS practices among our cohort, used by 48.4% (n = 148) and 59.2% (n = 181) of participants respectively. Since roughly half of the participants (n = 165, 47,1%) stated that clinical research was their primary focus, this may explain why a similar proportion report preregistering their study protocols, which is a World Health Organization requirement to conduct clinical trials [45]. However, barriers such as not knowing how to create a study registration and not knowing where to find relevant reporting guidelines were indicated by participants, once again suggesting that further support, training, and knowledge of the OS practices may aid their implementation in our participants' research.

## Strengths and limitations

In this study, we leveraged a cross-sectional survey due to its efficient and cost-effective nature, enabling us to capture a snapshot of our target group without requiring long-term follow-up. This approach also enabled us to make broader generalizations across CAIM researchers worldwide, which is another notable strength. To achieve this, we surveyed a sample of CAIM researchers who had varying experiences and perceived barriers related to open science. We also achieved an excellent completion rate among those who responded. However, we acknowledge that our study has limitations such as our low response rate and offering the survey solely in English, likely making it more difficult for non-English speakers to participate and share their thoughts and opinions. As a result, our findings may not be applicable to researchers who engage in open science practices using languages other than English. Furthermore, our cross-sectional survey methodology is inherently limited by recall and non-response biases. We also underestimated the response rate of our survey since we did not account for invalid or non-functioning email addresses (e.g., bounce-backs after sending) or authors with autoreplies indicating vacation or sick leave.

## Conclusion

In this study, our goal was to explore what are CAIM researchers' practices and perceived barriers related to OS. The survey participants shared their experiences, thoughts, and attitudes, which can provide valuable insights for both the open science and CAIM communities. Although participants were familiar with and implemented open science practices such as publishing open access, registering a protocol, and using reporting guidelines, funding barriers and concerns about sharing their work before publication persisted. To the best of our knowledge, this is the first study to document how CAIM researchers view and adopt open science practices in their scholarly work. While previous literature has suggested solutions to increase the adoption of open science in other scientific disciplines, our study provides a solid foundation for implementing current strategies and developing new approaches to enhance the uptake of open science in CAIM. Future work can build upon our baseline study to monitor the implementation of open science interventions in the CAIM field. We hope that our findings and analysis can also benefit other disciplines and contribute to the global adoption of open science, while avoiding the drawbacks of "closed research".

## Supporting information

**S1 File.**
(DOCX)

## Author Contributions

**Conceptualization:** Jeremy Y. Ng.

**Data curation:** Jeremy Y. Ng, Lucas J. Santoro.

**Formal analysis:** Jeremy Y. Ng, Lucas J. Santoro, Holger Cramer.

**Investigation:** Jeremy Y. Ng, Lucas J. Santoro, Kelly D. Cobey, Amie Steel, Holger Cramer, David Moher.

**Methodology:** Jeremy Y. Ng, Kelly D. Cobey, Amie Steel, Holger Cramer, David Moher.

**Project administration:** Jeremy Y. Ng.

**Supervision:** Jeremy Y. Ng, David Moher.

**Writing – original draft:** Jeremy Y. Ng, Lucas J. Santoro.

**Writing – review & editing:** Kelly D. Cobey, Amie Steel, Holger Cramer, David Moher.

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
