## [Decision Letter · Decision Letter 0]

2 Jan 2024

PONE-D-23-35865Complementary, Alternative, and Integrative Medicine Researchers’ Practices and Perceived Barriers Related to Open Science: An International, Cross-Sectional SurveyPLOS ONE

Dear Dr. Ng,

Thank you for submitting your manuscript to PLOS ONE. After careful consideration, we feel that it has merit but does not fully meet PLOS ONE’s publication criteria as it currently stands. Therefore, we invite you to submit a revised version of the manuscript that addresses the points raised during the review process.

We look forward to receiving your revised manuscript.

Kind regards,

Vincent Antonio Traag, Ph.D.

Academic Editor

PLOS ONE

Journal Requirements:

Additional Editor Comments:

Please make sure that the manuscript is consistent throughout the text. As the reviewers indicate, this is particularly relevant for the description of the sampling design. Please also make sure it is consistent with your actual sampling design (i.e. not a random sample). In addition, please clarify how the qualitative analysis is done. As the reviewers recommend, you may consider analysing the relationships between the various variables, but restricting to a more descriptive analysis is also acceptable.

Reviewers' comments:

Reviewer's Responses to Questions

**Comments to the Author**

1. Is the manuscript technically sound, and do the data support the conclusions?

Reviewer #1: Yes

Reviewer #2: Partly

Reviewer #3: Yes

2. Has the statistical analysis been performed appropriately and rigorously? 

Reviewer #1: Yes

Reviewer #2: N/A

Reviewer #3: I Don't Know

3. Have the authors made all data underlying the findings in their manuscript fully available?

Reviewer #1: Yes

Reviewer #2: Yes

Reviewer #3: Yes

4. Is the manuscript presented in an intelligible fashion and written in standard English?

Reviewer #1: Yes

Reviewer #2: Yes

Reviewer #3: Yes

5. Review Comments to the Author

Reviewer #1: The paper reports descriptive results of a survey of researchers in one research area in medicine (Complementary, Alternative, and Integrative Medicine) about their understanding and practices related to open science. The study has been well-designed and executed and the presentation is good. The data and the questionnaire have been made available publicly. The findings are interesting and can inform future initiatives to promote and advance open science in this area. I can't see any major error in the paper (just one minor) that needs to be fixed or any area for major improvement. However, I have a few points that authors might consider or respond to.

Twice it is mentioned that authors were randomly selected. I could not see in the research design where this random approach was deployed and the authors also say in the method that a convenience sample was used. This is contradictory. I believe the approach has been convenient sampling (all authors with an email published within a time frame in certain journals were invited and those who self-selected to participate, completed the survey). Therefore, the mention of 'random' should be removed from the paper.

A lot of demographic questions have been asked without making any use of them. I understand some of them provide contextual information, but generally, the best practice is not to ask a question if the information is not to be used. Not sure what value knowing about minorities or visibility adds here. Too late to change the questionnaire, but beyond providing context, have the authors explored or want to explore any demographic differences in practising or awareness of open science (e.g. are younger researchers more likely to practice OS)? The same can be done for different questions if authors want to explore their data any further (for instance did having any training in OS make a difference in practising it?) I understand if the authors decide to keep the results descriptive as is.

I believe a bit more literature can provide a good context for some of the findings. The authors can compare some of the results with studies in other fields (e.g. many other studies have found fees as the main barrier to OA publishing). Moreover, I am wondering if more information about CAIM field might be helpful in understanding the results. The percentage of registering a protocol for instance is remarkably high (48%), maybe this is the norm or a requirement (by funders...). So any more information about this field might better explain the findings.

Reviewer #2: The authors mention that they make a qualitative analysis of the open-ended responses, but it is not clear from the text.

In the methods it is said that the sample is random, but it is not.

The data on age, gender and professional group to which they belong is only visible in the raw data.

Has any analysis of the reliability or consistency between some of the answers been done?

The paper does not make any statistical analysis of the data, it only gives percentages, it would have been interesting to see the crosstabs between e.g. gender, age, and the responses to the survey.

In the tables of the survey raw data, the number of responses is given with a decimal place in front, what does this figure mean?

The work confirms what has already been seen in other disciplines but is somewhat weak because of the poor treatment of the data.

Reviewer #3: General:

1. Authors should mention that a preprint version of this study is available on medRxiv 2023.10.24.23297458; doi: https://doi.org/10.1101/2023.10.24.23297458 .

Abstract:

1. Please note that the study objectives presented in the abstract are not in alignment with those mentioned in other sections.

2. Future research mentioned here - such as increasing CAIM researchers’ funding and educational resources to implement OS practices – is not aligned with what is mentioned in the conclusions of the manuscript, where you stated that - Future work can build upon our baseline study to monitor the implementation of open science interventions in the CAIM field. Please consider to align it with the conclusions presented at the end of the manuscript.

Introduction:

1. As presented in your results, the authors consider that OA publication is very expensive and that more funding is needed to promote its adoption. So, in the literature review, it would be interesting to mention that there are several OA routes and that not all imply charges to readers or authors.

2. When you mentioned - By limiting constraints on information access, the OS movement also supports equity - it would be interesting to not limit equity to the user's perspective. Equity also involves reducing or eliminating the barriers to the publishing system that authors face. Those from lower-income countries, for example, cannot pay for open access costs.

3. You mentioned - Knowing about CAIM researcher’s opinions of OS and the barriers they face may benefit the adoption of CAIM by the public, the further implementation of evidence-based practices in CAIM research, and the removal of stigmas regarding the unreliability and untrustworthiness of CAIM20. – Considering your study goals and results, you can not assume that your study will bring those benefits. Those are the benefits that OA can bring to health and medical research in general as often reported in the literature. As you mention - These surveys serve as the foundation for developing new interventions that more successfully implement open research (…). So, by exploring what are CAIM researchers’ practices and perceived barriers related to OS, you are only able to better understand the experiences of those authors and identify the need for further work to promote the adoption of OA practices in CAIM research.

4. Please make sure that the goals listed here are in alignment with those in the abstract, discussion and conclusion – here you mention – Our goal was to explore (…), in the abstract and discussion – Our objective was to access, and in the conclusion you – In this study, we explored (…). Please be consistent

Methods:

1. Study Design - Please make sure that the selected journals mentioned here – CAIM journals – are in alignment with those mentioned in the introduction section – journals indexed in Scopus.

2. Survey design - When you mentioned – The majority of the survey questions were multiple choice – please provide more detailed information: How many? What other types of questions were included? How questions were grouped? The questions were mandatory or optional? (…)

3. In the Data Management, Analysis, and Reporting section, you mentioned – in respect to qualitative data (…) open-ended questions. Please make sure that the information mentioned here is aligned with the one mentioned in the Survey Design section, where no information regarding open-ended questions was mentioned. Please add more details on how qualitative data was analyzed and by whom.

Results:

1. In the Methods Section, you mentioned that the authors were selected at random, but here you say that all those who had an email available were contacted. You should clarify, otherwise it seems contradictory.

2. As the survey had open-ended options, please report how many respondents you had per each such question, how were they analyzed/coded and how many different categories you found for those answers.

3. Please avoid using less, many, most or similar expressions without data like you did In the Using Reporting Guidelines section, where you mentioned - 1) many respondents report not (…) please specify n and %.

Discussion:

1. You stated that – These results are consistent with previous studies – but then you only cite one study (ref. 32). Please consider enriching your literature review. Other studies besides Pardo Martínez C and Poveda A, like this recent one, https://onlinelibrary.wiley.com/doi/10.1111/gtc.13015, reporting the results of a questionnaire survey of MBSJ members about preprints and open access publishing, could be of interest for the discussion and likely could be included in the literature review.

2. As mentioned before, when mentioning many or the majority (…) please add n and %.

Conclusion:

1. As mentioned before, when mentioning many or the majority (…) please add n and %. Also, make sure that the goals listed here are in alignment with those mentioned before – goals or objectives / explore or access.

2. As mentioned before future research mentioned here is not aligned with what is mentioned in the Abstract. We suggest you keep what you mention here - Future work can build upon our baseline study to monitor the implementation of open science interventions in the CAIM field - assuring a stronger final message, and not limiting OA to funding or education.

6. PLOS authors have the option to publish the peer review history of their article (what does this mean?). If published, this will include your full peer review and any attached files.

Reviewer #1: **Yes: **Hamid R. Jamali, Charles Sturt University

Reviewer #2: **Yes: **Remedios Melero

Reviewer #3: No

---

## [Author Response · Author response to Decision Letter 0]

2 Feb 2024

Response to Reviewers

Dear Editor:

Thank you for your email. I am pleased to hear that you believe that our manuscript will have the potential to be published in PLOS ONE following requested revisions.

As per your request, please find our response to the peer-reviewers’ comments with every change outlined point by point below:

Journal Requirements:

• We kindly thank the editor for providing their feedback on our manuscript.

• We have ensured the manuscript meets PLOS ONE’s style requirements.

• We have moved the ethics statement to the Methods which includes all consent information,

• We have ensured the ORCID iD is validated in the editorial manager.

• We have moved the ethics statement to the methods section.

• We have included captions for the supporting information files. 

• We have ensured the reference list is complete and correct.

Additional Editor Comments:

Please make sure that the manuscript is consistent throughout the text. As the reviewers indicate, this is particularly relevant for the description of the sampling design. Please also make sure it is consistent with your actual sampling design (i.e. not a random sample). In addition, please clarify how the qualitative analysis is done. As the reviewers recommend, you may consider analysing the relationships between the various variables, but restricting to a more descriptive analysis is also acceptable.

• We have removed the mention of the term ‘random’ when describing our sample and we have also updated the description of the qualitative analysis. Given that our data is shared openly and available to anyone looking to analyze our data further, we have restricted our discussion in our manuscript to focus on a descriptive analysis only.

Reviewers' comments:

Reviewer's Responses to Questions

Comments to the Author

1. Is the manuscript technically sound, and do the data support the conclusions?

Reviewer #1: Yes

Reviewer #2: Partly

Reviewer #3: Yes

2. Has the statistical analysis been performed appropriately and rigorously?

Reviewer #1: Yes

Reviewer #2: N/A

Reviewer #3: I Don't Know

3. Have the authors made all data underlying the findings in their manuscript fully available?

Reviewer #1: Yes

Reviewer #2: Yes

Reviewer #3: Yes

4. Is the manuscript presented in an intelligible fashion and written in standard English?

Reviewer #1: Yes

Reviewer #2: Yes

Reviewer #3: Yes

5. Review Comments to the Author

Reviewer #1: The paper reports descriptive results of a survey of researchers in one research area in medicine (Complementary, Alternative, and Integrative Medicine) about their understanding and practices related to open science. The study has been well-designed and executed and the presentation is good. The data and the questionnaire have been made available publicly. The findings are interesting and can inform future initiatives to promote and advance open science in this area. I can't see any major error in the paper (just one minor) that needs to be fixed or any area for major improvement. However, I have a few points that authors might consider or respond to.

• We kindly thank this reviewer for providing their feedback on our manuscript.

Twice it is mentioned that authors were randomly selected. I could not see in the research design where this random approach was deployed and the authors also say in the method that a convenience sample was used. This is contradictory. I believe the approach has been convenient sampling (all authors with an email published within a time frame in certain journals were invited and those who self-selected to participate, completed the survey). Therefore, the mention of 'random' should be removed from the paper.

• Thank you for catching this. We have removed the mention of ‘random’ from the paper.

A lot of demographic questions have been asked without making any use of them. I understand some of them provide contextual information, but generally, the best practice is not to ask a question if the information is not to be used. Not sure what value knowing about minorities or visibility adds here. Too late to change the questionnaire, but beyond providing context, have the authors explored or want to explore any demographic differences in practising or awareness of open science (e.g. are younger researchers more likely to practice OS)? The same can be done for different questions if authors want to explore their data any further (for instance did having any training in OS make a difference in practising it?) I understand if the authors decide to keep the results descriptive as is.

We had originally decided to keep the results descriptive because we have shared all of our data openly. That being said, based on this reviewer’s feedback, we have further added 6 crosstab files to OSF which provide stratifications by key demographic questions including gender, age, minority group, caregiver, region, and career stage.

I believe a bit more literature can provide a good context for some of the findings. The authors can compare some of the results with studies in other fields (e.g. many other studies have found fees as the main barrier to OA publishing). Moreover, I am wondering if more information about CAIM field might be helpful in understanding the results. The percentage of registering a protocol for instance is remarkably high (48%), maybe this is the norm or a requirement (by funders...). So any more information about this field might better explain the findings.

• We have added some additional comparative literature from other disciplines/fields which now reads “Comparative studies such as the survey of Molecular Biology Society of Japan members found that 76.6% (n = 478) respondents have published open access35,41,42. Additionally, 91.5% (n = 571) of respondents in the same survey reported wanting to publish open access35. Costs, however, remain a barrier to those wishing to publish open access, with some researchers having to use their personal funds towards APCs35. Ideally, OS practices such as open access publishing promote equity and reduce barriers that end-users and authors face. Tools such as Sherpa Romeo exist to help users understand individual journal policies regarding preprinting, post-printing, and open access publishing. Finding journals that lack APCs or imply charges should therefore be easier for readers and authors, which is especially important for those in lower-income countries.” However, studies looking at open science concerning CAIM are quite sparse. For example, we have provided additional context for the percentage of registering a protocol at the end of the discussion. 

Reviewer #2: The authors mention that they make a qualitative analysis of the open-ended responses, but it is not clear from the text.

• We kindly thank this reviewer for providing their feedback on our manuscript. We have opted to keep the results of our study solely descriptive, as mentioned by the editor, and also because we have shared all of our data openly. With regards to the open-ended questions, all of the qualitative data was descriptively categorized. We have bolstered our methods section “Data Management, Analysis, and Reporting” to better describe this process.

In the methods it is said that the sample is random, but it is not.

• Thank you for catching this. We have removed the mention of ‘random’ when describing the sample.

The data on age, gender and professional group to which they belong is only visible in the raw data.

• We have represented this data in Table 3 of our manuscript.

Has any analysis of the reliability or consistency between some of the answers been done?

• We have not conducted an analysis concerning the reliability or consistency of our results. This has now been added as a limitation of our study. Our survey was also purpose built and future research could see to validate a tool to track these behaviors. 

The paper does not make any statistical analysis of the data, it only gives percentages, it would have been interesting to see the crosstabs between e.g. gender, age, and the responses to the survey.

• We had originally decided to keep the results descriptive because we have shared all of our data openly. That being said, based on this reviewer’s feedback, we have further added 6 crosstab files to OSF which provide stratifications by key demographic questions including gender, age, minority group, caregiver, region, and career stage.

In the tables of the survey raw data, the number of responses is given with a decimal place in front, what does this figure mean?

• We have edited the raw data to properly indicate the percentage of participants who answered the question. Previously, this number was expressed as a decimal. 

The work confirms what has already been seen in other disciplines but is somewhat weak because of the poor treatment of the data.

Reviewer #3: General:

• We kindly thank this reviewer for providing their feedback on our manuscript.

1. Authors should mention that a preprint version of this study is available on medRxiv 2023.10.24.23297458; doi: https://doi.org/10.1101/2023.10.24.23297458 .

• We have added the link to our preprint in the Methods Transparency Statement.

Abstract:

1. Please note that the study objectives presented in the abstract are not in alignment with those mentioned in other sections.

• We have revised our statement in the abstract “our objective was to assess practices and perceived barriers towards OS among CAIM researchers” so that it now closely aligns with the objective of our manuscript as well as the objectives mentioned in other areas of the manuscript.

2. Future research mentioned here - such as increasing CAIM researchers’ funding and educational resources to implement OS practices – is not aligned with what is mentioned in the conclusions of the manuscript, where you stated that - Future work can build upon our baseline study to monitor the implementation of open science interventions in the CAIM field. Please consider to align it with the conclusions presented at the end of the manuscript.

• We have changed the future work in the abstract to align with those mentioned in the conclusion.

Introduction:

1. As presented in your results, the authors consider that OA publication is very expensive and that more funding is needed to promote its adoption. So, in the literature review, it would be interesting to mention that there are several OA routes and that not all imply charges to readers or authors.

• We have noted preprinting, postprinting, as well as Sherpa Romeo where readers and authors can find detailed information regarding journal policies on these topics and APCs/costs. For example “Tools such as Sherpa Romeo exist to help users understand individual journal policies regarding preprinting, post-printing, and open access publishing. Finding journals that lack APCs or imply charges should therefore be easier for readers and authors, which is especially important for those in lower-income countries.”

2. When you mentioned - By limiting constraints on information access, the OS movement also supports equity - it would be interesting to not limit equity to the user's perspective. Equity also involves reducing or eliminating the barriers to the publishing system that authors face. Those from lower-income countries, for example, cannot pay for open access costs.

• We have noted the importance of lowering APCs and routes/tools such as preprinting and Sherpa Romeo to help authors and readers, especially those in lower-income countries. For example “Tools such as Sherpa Romeo exist to help users understand individual journal policies regarding preprinting, post-printing, and open access publishing. Finding journals that lack APCs or imply charges should therefore be easier for rea

---

## [Decision Letter · Decision Letter 1]

14 Mar 2024

Complementary, Alternative, and Integrative Medicine Researchers’ Practices and Perceived Barriers Related to Open Science: An International, Cross-Sectional Survey

PONE-D-23-35865R1

Dear Dr. Ng,

We’re pleased to inform you that your manuscript has been judged scientifically suitable for publication and will be formally accepted for publication once it meets all outstanding technical requirements.

Kind regards,

Vincent Antonio Traag, Ph.D.

Academic Editor

PLOS ONE

Additional Editor Comments (optional):

Reviewers' comments:

Reviewer's Responses to Questions

**Comments to the Author**

1. If the authors have adequately addressed your comments raised in a previous round of review and you feel that this manuscript is now acceptable for publication, you may indicate that here to bypass the “Comments to the Author” section, enter your conflict of interest statement in the “Confidential to Editor” section, and submit your "Accept" recommendation.

Reviewer #1: All comments have been addressed

Reviewer #2: All comments have been addressed

Reviewer #3: All comments have been addressed

2. Is the manuscript technically sound, and do the data support the conclusions?

Reviewer #1: Yes

Reviewer #2: Yes

Reviewer #3: Yes

3. Has the statistical analysis been performed appropriately and rigorously? 

Reviewer #1: Yes

Reviewer #2: (No Response)

Reviewer #3: N/A

4. Have the authors made all data underlying the findings in their manuscript fully available?

Reviewer #1: Yes

Reviewer #2: Yes

Reviewer #3: Yes

5. Is the manuscript presented in an intelligible fashion and written in standard English?

Reviewer #1: Yes

Reviewer #2: Yes

Reviewer #3: Yes

6. Review Comments to the Author

Reviewer #1: The authors have adequately responded to reviewers' comments and the revised manuscript is much improved. Further details in the method have added clarity, the context and discussion have been improved and further analysis provided in supplementary files will be useful for some readers.

Reviewer #2: I have realised that the auhtors have done and shared the crosstabs I suggested, however there is no mention within the text of potential conclusions, I mean, I have checked the crosstabs and found the frequencies and 3 more raws per question that are not identified, I wonder if they are adjusted residuals, or waht are they? if so, they could draw more conclusions, for instance the associations between dependent and independent variables

Reviewer #3: I thank the authors for paying attention to my comments and for addressing them all. The manuscript is now clearer, more consistent, coherent and of greater interest to readers.

7. PLOS authors have the option to publish the peer review history of their article (what does this mean?). If published, this will include your full peer review and any attached files.

Reviewer #1: **Yes: **Hamid R Jamali, Charles Sturt University

Reviewer #2: **Yes: **Remedios Melero

Reviewer #3: **Yes: **Susana Oliveira Henriques

---

## [Editor Report · Acceptance letter]

26 Apr 2024

PONE-D-23-35865R1 

PLOS ONE

Dear Dr. Ng, 

I'm pleased to inform you that your manuscript has been deemed suitable for publication in PLOS ONE. Congratulations! Your manuscript is now being handed over to our production team.

Kind regards, 

on behalf of

Dr. Vincent Antonio Traag 

Academic Editor

PLOS ONE